# LISTENER-AUGMENTED THINKING VLMS FOR ROBUST TEXT-TO-IMAGE ALIGNMENT

## ABSTRACT

Improving the generalization and robustness of reward models for visual preferences is crucial-especially under limited annotated data, where collection is costly and complex. While we show that reinforcement learning with verifiable rewards (RLVR) improves generalization, we identify a key failure mode: models often produce final answers that are inconsistent with their own chain-of-thought (CoT); when the same trace is shown to a frozen vision–language model (VLM), it frequently predicts a different choice, and accuracy drops as this inconsistency grows. To solve this, we propose a listener-augmented GRPO objective: a frozen listener re-evaluates the reasoner's CoT (with the final answer masked) and provides a dense soft reward, combined with an exact-match term for unbiased supervision. We further introduce a probability-based pairwise scoring scheme: at training time it yields calibrated confidence rewards for GRPO, and at inference time it supports an anchor-based strategy that can avoid $\mathcal{O}(n^2)$ comparisons while still producing reliable win probabilities. Under this framework, we establish a clear hierarchy - **single-image scoring** < **pairwise comparison** < **RL with CoT** < **RL with CoT + listener** - showing that conditioning on both images already boosts generalization, CoT improves further, and listener shaping provides the largest gains. Despite training on only 16% of HPDv2, our approach achieves 67.4% on the ImageReward test set and improves OOD accuracy by up to +6 pp on a modern 1.2M-vote benchmark, while reducing contradiction rates. Integrated into a Flow-GRPO text-to-image pipeline and evaluated with the latest HPSv3 metric on prompts from HPDv2 and T2I-CompBench, it yields consistent gains. We will release code and models to facilitate reproducibility.

## 1 INTRODUCTION

Optimizing vision–language models (VLMs) to robustly capture *human visual preferences* is a key problem for text-to-image and text-to-video generation Xu et al. (2023). Preference models determine both alignment with user intent and the ability to generalize beyond the training distribution. However, common approaches such as supervised fine-tuning (SFT) for reward modeling are limited: SFT tends to memorize Chu et al. (2025), and general reward models often lack robustness under distribution shift et al. (2022), motivating expensive annotation pipelines that still struggle to capture nuanced, subjective choices.

Reinforcement learning (RL) offers a more scalable paradigm for preference alignment. Recent systems (e.g., xAI Grok xAI (2025), OpenAI O1/O3 OpenAI (2024; 2025), Gemini 2.5 Pro Deep-Mind (2025), and DeepSeek DeepSeek (2024)) highlight RL's generalization benefits, particularly when paired with chain-of-thought (CoT) reasoning Xu et al. (2024a). In particular, Group Relative Policy Optimization (GRPO) improves stability via group-wise normalization and removes the value-network bottleneck DeepSeek (2024). *Nevertheless, we identify a systematic failure mode in naïve RL-trained preference reasoners: when generating CoT explanations, the same reasoning trace can lead the reasoner and an independent VLM to produce different final answers (Fig. 1). We refer to this as* listener disagreement*, and find that pairwise disagreement strongly correlates with accuracy drops, indicating that plausible reasoning alone does not guarantee consistent or correct decisions.*

Figure 1: Listener reduces reasoning–answer contradictions. Naive GRPO already generalizes well, but often produces correct answers misaligned with CoT traces. Listener shaping encourages CoT that supports the final answer and improves OOD accuracy.

To address this, we propose **listener-augmented GRPO** for preference reasoning. We extend the GRPO objective with a *listener-shaped soft reward*: a frozen VLM listener re-processes the reasoner's CoT (excluding the final answer token) and outputs the probability of the ground-truth choice via a binary softmax over "first/second" logits. This score is integrated into the RL signal, penalizing explanations that fail to convince an independent model and providing dense supervision without additional human labels. In addition to the listener mechanism, we analyze binary vs. pairwise vs. GRPO vs. listener variants and, leveraging the standardized JSON-style answer format of reasoning models, place *both* images in context and extract the final-step $\{\texttt{first}, \texttt{second}\}$ logits to obtain calibrated probabilities-used as soft signals during training and for lightweight anchor-based inference without enumerating all $\mathcal{O}(n^2)$ pairs.

We initialize our reward models with Qwen-2.5-VL-7B-Instruct Qwen (2024) and evaluate across in-domain (ImageReward) and challenging OOD data consisting of modern, high-quality generations. Critically, we restrict training to only **16%** of HPDv2. Even under this strict budget, our listener-augmented GRPO achieves **67.4%** on the ImageReward test set and surpasses strong GRPO/SFT baselines on the 1.2M-vote Rapidata-HSP benchmark Rapidata (2025), while reducing contradictory reasoning events. *Beyond proxy accuracy*, we also plug our rewards into the **Flow-GRPO** pipeline Liu et al. (2025) to fine-tune a real text-to-image generator, **Stable Diffusion 3.5-Medium** Stability (2024), and observe consistent gains when scored by the robust **HPSv3** metric Ma et al. (2025b) on HPDv2 Wu (2023) and T2I-COMPBENCHHuang et al. (2023) prompts. Under matched data, seeds, and schedules, our ablations further reveal a consistent *hierarchy* of preference modeling-**single-image (binary) < pairwise (two images jointly in context) < RL with CoT < listener-augmented RL with CoT**-observed on both in-domain and OOD evaluations. These results show that scaling to the full dataset is unnecessary for demonstrating strong generalization: our method already delivers state-of-the-art performance under limited supervision and calls into question the adequacy of training single-image preference predictors alone.

Our results show that listener-augmented RL improves the alignment of VLMs with binary, subjective human preferences - both in terms of proxy preference accuracy and end-to-end FlowMatching fine-tuning and offers a promising path for next-generation text-to-image and text-to-video systems.

**Contributions.**

- We empirically identify and quantify *listener disagreement* as a principal failure mode in RL-based visual preference modeling.
- We introduce a *listener-shaped soft reward* for GRPO that leverages an independent model's calibrated confidence over the reasoner's CoT to align explanations and decisions.

Figure 2: Each column uses the same prompts (from T2I-CompBench validation), *10 inference steps* during generation and the same training steps in a Flow-GRPO pipeline. Row-wise comparison across identical prompts. Row 1: baseline generations from Stable Diffusion 3.5-Medium (no RL). Row 2: images after fine-tuning the generator with Flow-GRPO using a GRPO reasoner as the reward. Row 3: images after Flow-GRPO using our Listener-GRPO reward.

- Under matched data (16% of HPDv2) and training steps, we establish a performance hierarchy: **binary (single-image)** < **pairwise** < **RL with CoT** < **listener-augmented RL**.
- We validate proxy accuracies on a standard ImageReward test set, large modern preference dataset Rapidata (2025) and in a Flow-GRPO Liu et al. (2025) T2I pipeline, showing superior accuracy and OOD generalization with smaller data fractions. Qualitative examples are shown on the Figure 2 and in Appendix A.2.

## 2 RELATED WORK

**Visual preference models.** Early reward models such as **ImageReward** (Xu et al., 2023) and **PickScore** (Kirstain et al., 2023) boost in-domain alignment yet struggle on modern generative outputs. **HPSv2** (Wu, 2023) improves robustness with a larger, bias–controlled dataset, but the collection process is costly and it's generalization is already limited. The recent **VisionReward** framework (Xu et al., 2024b) tackles generalization by annotating *multi-objective* sub-criteria (aesthetics, realism, prompt fidelity) and fine-tuning VLMs. While effective, its hierarchical markup pipeline further increases annotation complexity. Industry adoption therefore still seeks *light-weight* preference tuning strategies that generalize without human annotations.

**RL vs. SFT for alignment.** A growing evidence shows that reinforcement learning gets better generalization than supervised fine-tuning. Chu et al. (2025) demonstrate that SFT memorizes training distributions, whereas RL variants transfer better across tasks. The **DeepSeek** line adopts **Group Relative Policy Optimization** (GRPO) to scale reasoning models with minimal overhead (Shao et al., 2024; dee, 2024). GRPO's group-normalized update stabilizes learning without a value head, making it attractive for vision-language fine-tuning.

**Direct preference optimization.** **DPO** removes the need to fit a separate reward model by directly optimizing a contrastive preference objective (Rafailov et al., 2023). It has been successfully adapted

beyond language to diffusion and video generators: DIFFUSION-DPO significantly improves Stable Diffusion XL (Wallace et al., 2023), they report even better results when ranking images with a reward model; the 30B-parameter STEP-VIDEO T2V model applies *Video-DPO* to cut artifacts and lift FID/CLIPScores across a new video benchmark (*et al.*, 2025); HUNYUANVIDEO fine-tunes its 13 B text-to-video backbone with a preference-optimized loss (Kong et al., 2024); LTX-VIDEO reports a $+15\%$ preference swing after a lightweight DPO pass, while retaining real-time generation speed (HaCohen et al., 2024). These industry results highlight the demand for simple preference methods that scale without complex annotation pipelines-an objective our listener-shaped GRPO strategy fulfills while further improving reasoning consistency.

**Our contribution.** Building on the insight that RL generalizes better than SFT, we uncover a novel *listener disagreement* failure mode: VLM accuracy consistently declines as the distance between its predictions and those of an independent listener increases. To address this, we propose a listener-shaped GRPO objective that improves out-of-distribution generalization without requiring additional annotations or complex multi-objective markup-offering precisely the kind of simplicity needed for scalable T2I/T2V systems.

## 3 PRELIMINARIES

Vision-Language Models (VLMs) are large language models that take both text and image embeddings as input and generate text, enabling multimodal reasoning. Visual generative models, such as diffusion models (e.g., Stable Diffusion Stability (2024), DALL·E 3, Flux), synthesize images by iteratively denoising random noise, with text conditioning provided by text encoders, LLMs, or VLMs. Our work focuses on aligning the outputs of such generative models with human preferences using a VLM-based reasoner.

### 3.1 PREDICTING HUMAN PREFERENCES

Given a text prompt $P$ and a pair of images $I$, we learn a scoring function

$$f_\theta : (I, P) \longmapsto [0, 1],$$

where higher values indicate stronger human preference. For a set of images $\{I_1, \ldots, I_N\}$ associated with the same prompt, pairwise preference is inferred by score comparison; i.e. $I_a$ is preferred over $I_b$ if $f_\theta(I_a, P) > f_\theta(I_b, P)$. Modelling a *scalar* score instead of a direct binary label enables the soft reward scheme introduced in §4.

### 3.2 DATASETS

Benchmarking visual preference models requires careful curation of both training and evaluation datasets to ensure fair comparison, generalization assessment, and reproducibility. The protocol in this paper involves:

- **Training on established datasets:** Reward models are trained using widely adopted HPDv2. It contains large numbers of annotated image pairs, generated from a variety of older diffusion models (e.g., Stable Diffusion versions 1.x, 1.4/1.5).

- **Out-of-Distribution (OOD) evaluation:** Generalization is assessed using newer, more challenging datasets. For instance, RAPIDATA-HSP is constructed from the outputs of state-of-the-art generative models (e.g., DALL·E 3, Midjourney v6, Flux), which are not present in the training data, providing a robust test for OOD performance.

- **Evaluation with complex prompts:** To assess how reward models generalize to challenging compositional settings (e.g., geometry, shape, attribute binding), we additionally use prompts from T2I-COMPBENCH. These prompts are employed within the Flow-GRPO pipeline to evaluate improvements when training an actual text-to-image generator.

This protocol is motivated by the need to evaluate models under distribution shift, to prevent overfitting to older generative models and to provide large and robust benchmarking.

Table 1 summarises the specific datasets used in this work, detailing their scale, source models, and how they are utilized for training and evaluation.

Table 1: Datasets used for training and evaluation.. CompBench is used for Flow-GRPO pipeline (training and validation of a flow matching model), others are used for training and validation of reward models.

| Dataset | Scale | Source models | Split | Usage |
|---|---|---|---|---|
| ImageReward Xu et al. (2023) | 137 k pairs | SD 1.x | train / val / test | train/eval |
| HPSv2 Wu (2023) | 798 k pairs | SD 1.4/1.5 | train | train |
| Rapidata-HSPRapidata (2025) | 1.2 M votes | Modern | whole train (OOD) | eval |
| CompBenchHuang et al. (2023) | 14 k prompts | None | train / val | train/eval |

### 3.3 GROUP RELATIVE POLICY OPTIMISATION (GRPO)

We train the preference reasoner with *Group Relative Policy Optimisation* (GRPO) DeepSeek (2024), a variant of PPO that removes the value network by normalising rewards within a rollout group of size $G$. For each state $s$ we sample actions $\{a_i, r_i\}_{i=1}^{G}$ and compute a group-normalised[1] advantage

$$A_i = \frac{r_i - \mu}{\sigma + \varepsilon}, \qquad \mu = \tfrac{1}{G} \sum_i r_i, \ \sigma^2 = \tfrac{1}{G} \sum_i (r_i - \mu)^2. \tag{1}$$

The policy update maximises the clipped objective

$$\mathcal{L}(\theta) = \mathbb{E}_i\big[\min\big(\rho_i A_i,\ \mathrm{clip}(\rho_i, 1\pm\epsilon)\, A_i\big)\ -\ \lambda\, \mathrm{KL}\big(\pi_\theta(\cdot|s) \,\|\, \pi_{\mathrm{ref}}(\cdot|s)\big)\big] \tag{2}$$

where $\rho_i = \pi_\theta(a_i|s)/\pi_{\theta_{\mathrm{old}}}(a_i|s)$. GRPO's group-based normalisation stabilises training and eliminates the memory overhead of a separate value head.

In the following section, we describe the main failure mode observed when applying standard RL methods to visual preference reasoning, and introduce our proposed listener-shaped reward framework to address it.

## 4 METHOD: LISTENER–SHAPED REWARDS

**Method Overview.** Our approach couples two ingredients. First, *listener-augmented GRPO* (Sec. 4.4): a frozen listener re-evaluates the reasoner's CoT (with the final answer masked) and returns a dense, verifiable soft reward. We begin by quantifying the disagreement failure mode that motivates listener shaping (Sec. 4.1) and then explain why we prefer a soft, pre-answer reward (Sec. 4.2). Second, *two-image conditioning with structured probabilities* (Sec. 4.3): the reasoner sees both images jointly and emits a standardized JSON decision {first,second}, exposing calibrated, post-CoT logits that we also use for efficient anchor-based inference.

### 4.1 FAILURE MODE: LISTENER DISAGREEMENT

While GRPO improves generalization, we observe a reasoning-specific failure mode: *given the same chain-of-thought (CoT) trace*, the reasoner and an independent frozen VLM *listener* often produce different final predictions. We call this *listener disagreement*. Figure 3 shows that accuracy (even in-distribution) declines monotonically as the listener's and reasoner's score vectors diverge. Concretely, for each pair (winner, loser), we obtain the reasoner's probability scores $(p_{\mathrm{win}}^R, p_{\mathrm{lose}}^R)$ and the listener's scores $(p_{\mathrm{win}}^L, p_{\mathrm{lose}}^L)$ given the same CoT, and measure their $L_2$ distance-revealing that robust alignment requires explanations that are *mutually consistent* across models.

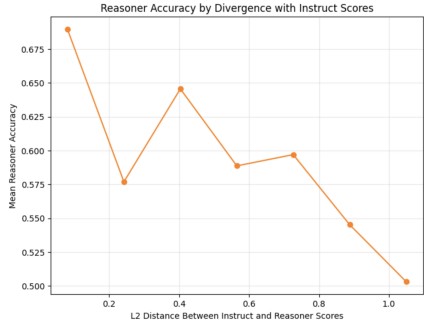

Figure 3: **Listener–reasoner disagreement.**

---

[1]We use $G = 10$.

## 4.2 WHY SOFT LISTENER REWARDS?

Hard decisions (keep/discard based on a threshold) waste signal and create extra hyperparameters. Instead, we use a *soft* reward from the listener for three reasons:

- **No example is thrown away.** Ambiguous or close cases still provide graded credit, improving data efficiency under our data budget.
- **No threshold tuning.** We avoid hand-tuned disagreement cutoffs and the instability that comes from crossing a hard boundary.
- **Better credit assignment.** A continuous signal reflects how convincing the CoT is to an independent model, which GRPO's group-normalization turns into stable advantages. Importantly, the reward is computed at the *pre-answer* step with the final token masked, preventing leakage while still evaluating the persuasiveness of the trace.

## 4.3 TWO-IMAGE CONDITIONING WITH STRUCTURED PROBABILITIES

We condition the VLM on *both* images jointly and ask for a structured, two-token decision in a standardized JSON-style format: `<answer>{"preferred":"first|second"}</answer>`. This has two benefits. First, direct comparison is less sensitive to absolute calibration drift across generators than single-image scoring and intuitively should overfit less, because the reward model seeks *relative* improvements. Second, because the answer space is $\{\texttt{first}, \texttt{second}\}$, we can read the model's next-token logits at the final step to obtain calibrated decision probabilities *after* the CoT.

Concretely, for the *reasoner* we extract logits $f_{\texttt{first}}^{\text{R}}, f_{\texttt{second}}^{\text{R}}$ and define

$$p_{\text{win}}^{\text{R}} = \text{softmax}\big([f_{\texttt{first}}^{\text{R}}, f_{\texttt{second}}^{\text{R}}]\big)[y^{\star}].$$

For the *listener* we analogously obtain $p_{\text{win}}^{\text{L}}$ from its two-token logits, computed on the *same* CoT with the final answer masked (used in Eq. 3). These structured probabilities supply dense, calibrated signals without introducing an auxiliary scoring head. To mitigate position bias, we uniformly randomize left/right image order during training.

**Anchor-based inference.** When ranking $n>2$ candidates, we avoid $\mathcal{O}(n^2)$ pairings by *anchor sampling*: compare each $I_k$ against a small set of anchors $\{I_a\}$ and average the resulting win probabilities $\hat{s}(I_k) = \frac{1}{|\mathcal{A}|} \sum_{a \in \mathcal{A}} p_{\text{win}}^{\text{R}}(I_k \text{ vs. } I_a)$. This yields reliable scores with less complexity and, in practice, improves stability as anchors increase (see Sec. 5; ablation in Sec. 5.4.1).

## 4.4 LISTENER-AUGMENTED GRPO

Let $C = \{V, P, T, A_{<t}\}$ denote the context (visual input $V$, prompt $P$, CoT tokens $T$, and the partial answer $A_{<t}$). At the time step *right before* the final answer token, the listener produces next-token logits for the two answer tokens, $f_{\texttt{first}}$ and $f_{\texttt{second}}$. We mask the true answer token from the listener to prevent leakage and compute a binary softmax:

$$p_{\text{win}} = \text{softmax}\big([f_{\texttt{first}}, f_{\texttt{second}}]\big)[y^{\star}],$$

the listener's probability of the ground-truth winner $y^{\star}$ given the reasoner's CoT. We combine this with a formatting check $r_{\text{fmt}} \in \{0, 1\}$ and a strict exact-match term $r_{\text{acc}} = \mathbb{I}[a = y^{\star}]$ to form the total reward used by GRPO:

$$r = r_{\text{fmt}} + 0.5\, r_{\text{acc}} + 0.5\, p_{\text{win}}. \tag{3}$$

This reward supplies dense, calibrated supervision that *aligns the explanation with the decision* without discarding uncertain examples or relying on brittle thresholds. The listener is *frozen* throughout training; only the reasoner is updated. GRPO then computes group-normalized advantages and applies the standard clipped objective (Section 3.3).

## 5 EXPERIMENTS

### 5.1 EVALUATION TRACKS

We evaluate along two complementary stages:

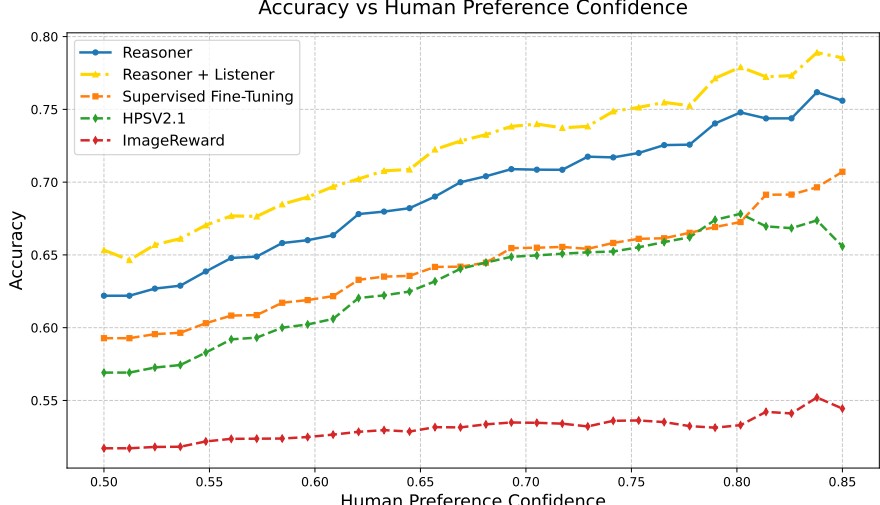

Figure 4: **Accuracy vs. human agreement** on the modern OOD dataset Rapidata (2025). Listener shaping improves accuracy across confidence bins.

1. **Proxy accuracy.** Mean accuracy on (i) the in-distribution ImageReward test split and (ii) a large, modern OOD dataset (Rapidata-HSP; 1.2M votes). For Rapidata-HSP we report *accuracy at confidence* by binning pairs by human-agreement threshold (Fig. 4).

2. **End-to-end generation.** We plug each reward into a **Flow-GRPO** pipeline to fine-tune **Stable Diffusion 3.5-Medium** Stability (2024). We train on **HPDv2** and **T2I-CompBench** prompts and evaluate on **HPDv2 test** prompts and **T2I-CompBench val** prompts. Generations are scored with the latest robust **HPSv3** Ma et al. (2025b). To avoid evaluator overlap and reward hacking, we do *not* use HPSv2 or ImageReward as evaluators for generations.

## 5.2 MODELS AND TRAINING SETUP

All reward models share identical initialization and data:

- **Backbone.** Qwen2.5-VL-7B-Instruct for both reasoner and listener.
- **Data budget. 16% of HPSv2**, with identical seeds/schedules across methods.
- **Methods compared.** *Binary* (single image, classical reward-model training), *Pairwise* (two images in context; predict `first`/`second`), *GRPO (CoT)*, and *Listener-GRPO (CoT)*.
- **Listener reward.** $r = r_{\text{fmt}} + 0.5\,r_{\text{acc}} + 0.5\,p_{\text{win}}$, where $p_{\text{win}}$ is the listener's probability of the ground-truth winner given the reasoner's CoT (final answer masked). The listener is *frozen*; only the reasoner is updated.
- **Order randomization.** Left/right image order is uniformly randomized during training; at inference we aggregate over randomly sampled anchors (Sec. 5.4.1).
- **Training hyperparameters.** GRPO on $8\times$H100; learning rate $1 \times 10^{-6}$; batch size 1 with 4 grad-acc steps; sequence length 512; temperature 1.1 during training; group size $G{=}10$ rollouts. Inference uses **greedy** decoding.

## 5.3 MAIN RESULTS

### 5.3.1 PROXY ACCURACY (IN-DISTRIBUTION AND OOD)

Across identical data budgets (16% HPDv2), GRPO (CoT) improves over classical reward models; adding the listener yields a further (small but consistent) gain in-distribution and larger gains OOD. Table 2 reports ImageReward test accuracy; OOD improvements are shown via accuracy-at-confidence on Rapidata-HSP (Fig. 4).

| Method | Accuracy (%) |
|---|---|
| Single Human Wu (2023) | 65.3 |
| HPSv2.1 Wu (2023) | 66.8 |
| ImageReward Xu et al. (2023) | 65.1 |
| SFT on HPSv2 | 66.9 |
| GRPO (CoT) on HPDv2 | **67.2** |
| + Listener (ours) | **67.4** |

Table 2: ImageReward test accuracy. GRPO (CoT) surpasses classical reward models; listener shaping yields a further gain.

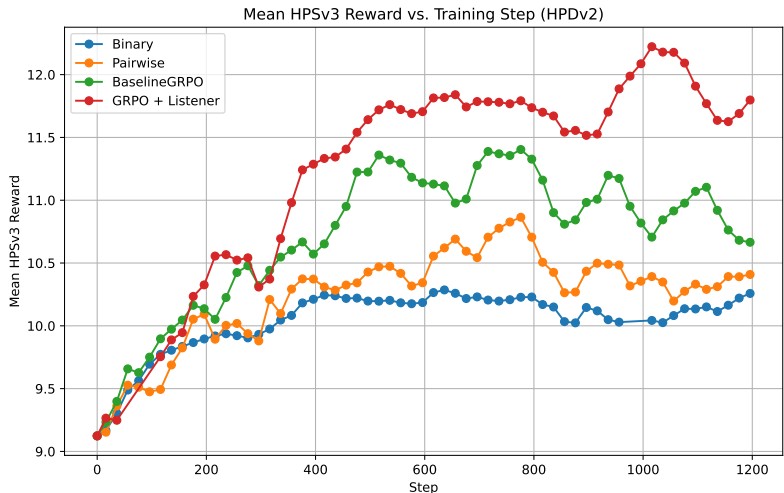

Figure 5: **Flow-GRPO on HPDv2 prompts.** HPSv3 reward measured on HPDv2 test prompts. Listener-GRPO yields stronger reward dynamics and preserves the hierarchy.

### 5.3.2 END-TO-END GENERATION (FLOW-GRPO)

We integrate each reward into Flow-GRPO and fine-tune **SD-3.5-Medium**. Relative to the reference setup, we (i) reduce images per prompt to **12** and (ii) halve the learning rate for stability. We train on **HPDv2** and **T2I-CompBench** prompts; we evaluate on **HPDv2 test** (400 prompts) and **T2I-CompBench val** (25 per category; 925 total). All end-to-end scores use **HPSv3**. Reward dynamics show a clear hierarchy and stronger gains for Listener-GRPO (Fig. 5, Fig. 6).

### 5.4 ABLATIONS

#### 5.4.1 INFERENCE STRATEGY: ANCHOR SAMPLING

For prompts with $n$ images, we avoid $\mathcal{O}(n^2)$ pairwise comparisons by using *anchor sampling*: pick an anchor $I_{\text{anc}}$ and compare each $I_k$ to the anchor in a single pass; aggregate across multiple anchors. Unless stated, we use **10 anchors** for all pairwise/GRPO/Listener-GRPO mod-

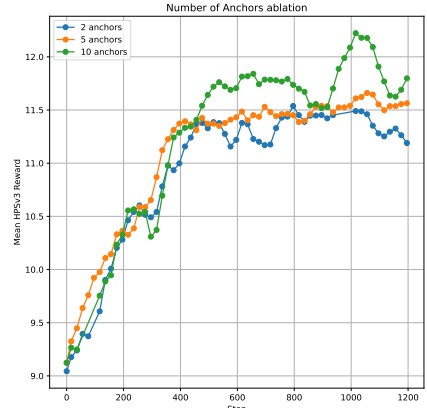

Figure 7: **Anchor count ablation.** More anchors improve quality.

els. An ablation (Fig. 7) shows how the end-to-end HPSv3 reward changes with 2, 5, or 10 anchors - more anchors improve stability/quality with diminishing returns.

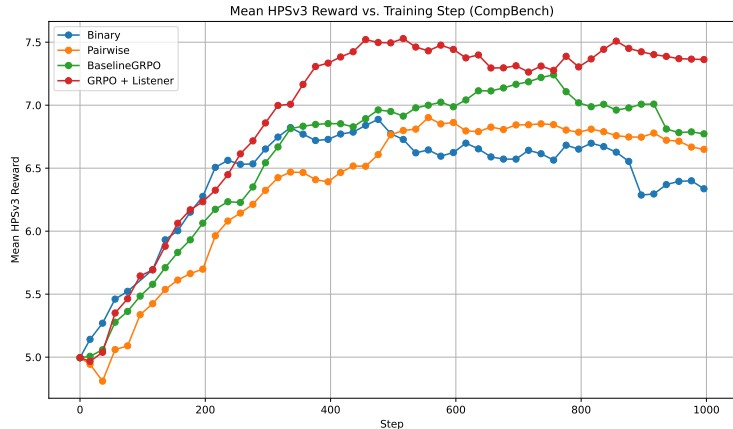

Figure 6: **Flow-GRPO on CompBench prompts.** HPSv3 reward measured on CompBench val prompts. Listener-GRPO remains strongest; the hierarchy is visible on complex compositional cases.

### 5.4.2 CONTRADICTION RATES

We quantify reasoning–answer contradictions using **Qwen2.5-14B-Instruct** as an evaluation-only judge. Listener-GRPO reduces contradictions (Table 3).

Table 3: Contradiction rate on high-confidence OOD pairs ($\geq$20 total votes; winner $\geq$ 80%). Judge: Qwen2.5-14B-Instruct.

| Model | Contradictions (%) |
|---|---|
| GRPO (CoT) | 10.1 |
| + Listener (ours) | **8.3** |

### 5.4.3 DO REASONING TRACES MATTER?

Following Ma et al. (2025a), we replace the CoT with the fixed string "I have finished thinking." Accuracy stays similar for GRPO but drops for Listener-GRPO (Table 4), indicating the listener shaping *uses* the trace rather than ignoring it. (Filtered high-confidence Rapidata pairs.)

Table 4: Effect of removing the CoT ("I have finished thinking"). Listener-GRPO relies on the trace to generalize.

| Model | Accuracy (%) |
|---|---|
| GRPO (CoT) | 72 |
| GRPO (no CoT) | 73 |
| + Listener (CoT) | **76** |
| + Listener (no CoT) | **70** |

## 6 CONCLUSION

We introduced *listener–augmented GRPO*, a compact recipe for **thinking VLMs** that converts reasoner–listener disagreement into a dense reward over CoT traces. Our two–image (pairwise) conditioning with a structured JSON answer exposes post-CoT logits; combined with listener shaping, this aligns explanations with decisions and improves robustness. An anchor-based inference can further avoid $\mathcal{O}(n^2)$ comparisons. Under a strict budget (**16%** HPDv2), the method reaches **67.4%** on ImageReward, improves OOD accuracy by up to **+6 pp** on a 1.2M-vote benchmark, and reduces reasoning–answer contradictions. As a reward for Flow-GRPO (SD-3.5-Medium), it yields consistent **HPSv3** gains with a better generalization to modern generators. Finally, we establish a hierarchy **single-image** $<$ **pairwise** $<$ **RL+CoT** $<$ **RL+CoT+listener** and show that pairing a thinking VLM with a frozen listener is a simple, scalable route to robust text-to-image alignment.

REPRODUCIBILITY STATEMENT

All experiments are reproducible: we fix random seeds, report datasets and hyperparameters, and will release anonymized code and training scripts with the submission.

ETHICS STATEMENT

Our study trains thinking VLM reward models for pairwise image preference and uses them to fine-tune text-to-image generators. We use publicly released datasets with stated licenses and anonymized human votes (HPDv2/HPSv2, ImageReward, Rapidata-HSP; details in §Data). We obtained no new human subjects data. Potential risks include (i) bias amplification toward specific aesthetics or demographics; (ii) misuse of preference models to optimize harmful or NSFW content; (iii) homogenization of creative outputs; and (iv) environmental impact from compute. Mitigations: we randomize image order; report OOD behavior; measure and reduce reasoning–answer contradictions; avoid training/evaluating on the same reward to reduce reward hacking; filter unsafe prompts in end-to-end experiments; and release code with a use policy and flags for content filtering. We will document known biases, hyperparameters, and seeds.

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

## A    APPENDIX

### USE OF LLMS

In accordance with ICLR 2026's policies on large language model usage, we disclose that we used LLMs only for minor text polishing and grammar fixes, not for generating content, claims, experiments, or analyses. All technical content, experiments, reasoning, and results were authored and vetted manually by the human authors. We take full responsibility for any errors or misstatements, including those introduced during polishing.

## A.1 PROMPTING DETAILS (FOR REPRODUCIBILITY)

To ensure comparability, all methods use identical prompts. **System prompt:**

> "The user has two images and a textual prompt. You need to reason carefully and produce an answer with reasoning in <think>...</think> where you should choose the best image."

**User prompt:**

```
f"User prompt: {prompt} Which image is better
given the prompt?  Analyze aesthetics, composition,
prompt alignment and other factors.  Provide your
reasoning in <think>...</think> tags and the final JSON
answer in <answer>{"preferred":"second"}</answer> or
{"preferred":"first"}."
```

## A.2 QUALITATIVE EXAMPLES

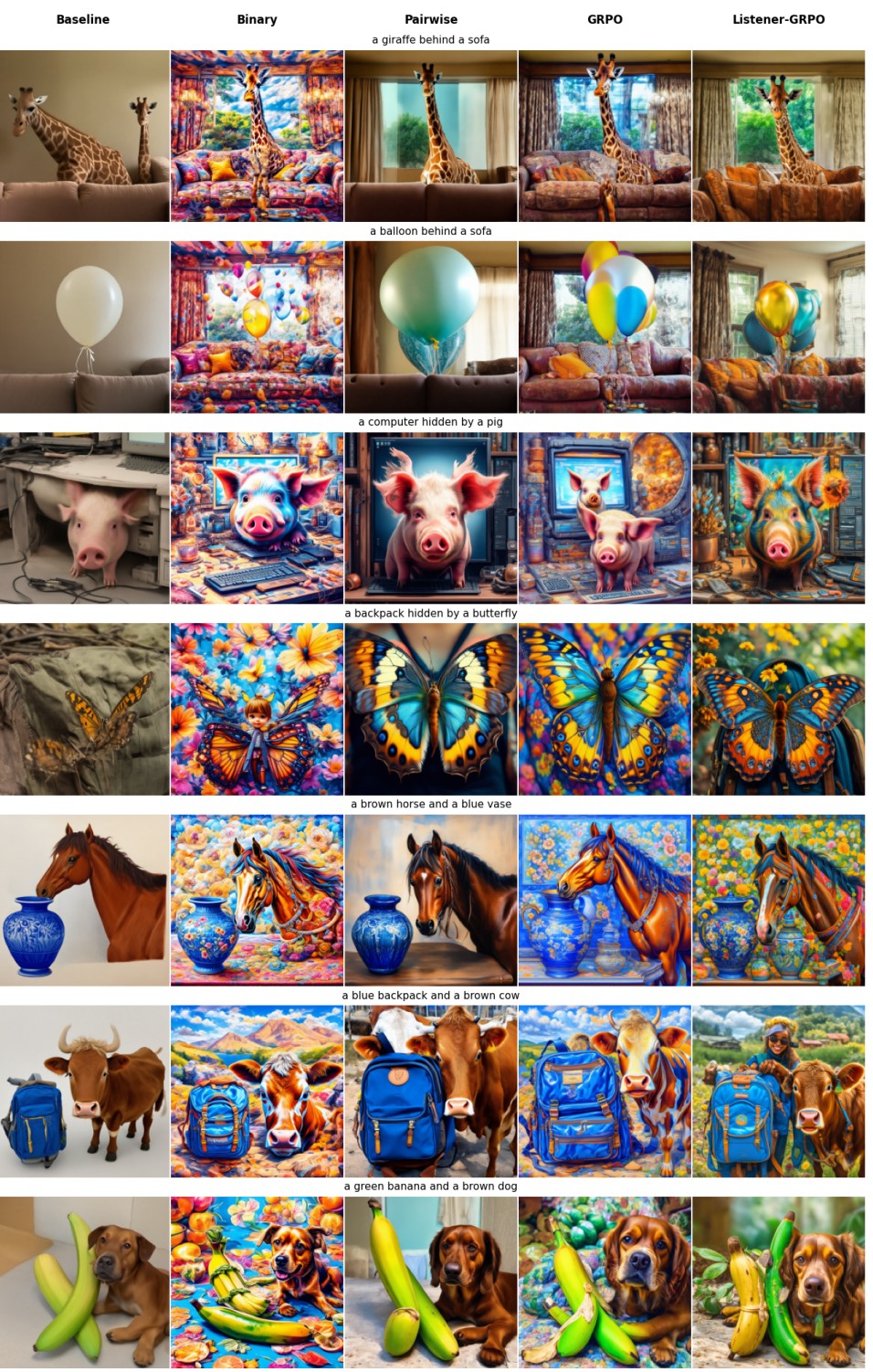

Figure 8: Qualitative comparison of **Stable Diffusion 3.5-Medium** generations. All checkpoints were trained with identical hyperparameters and evaluated with 10 inference steps.

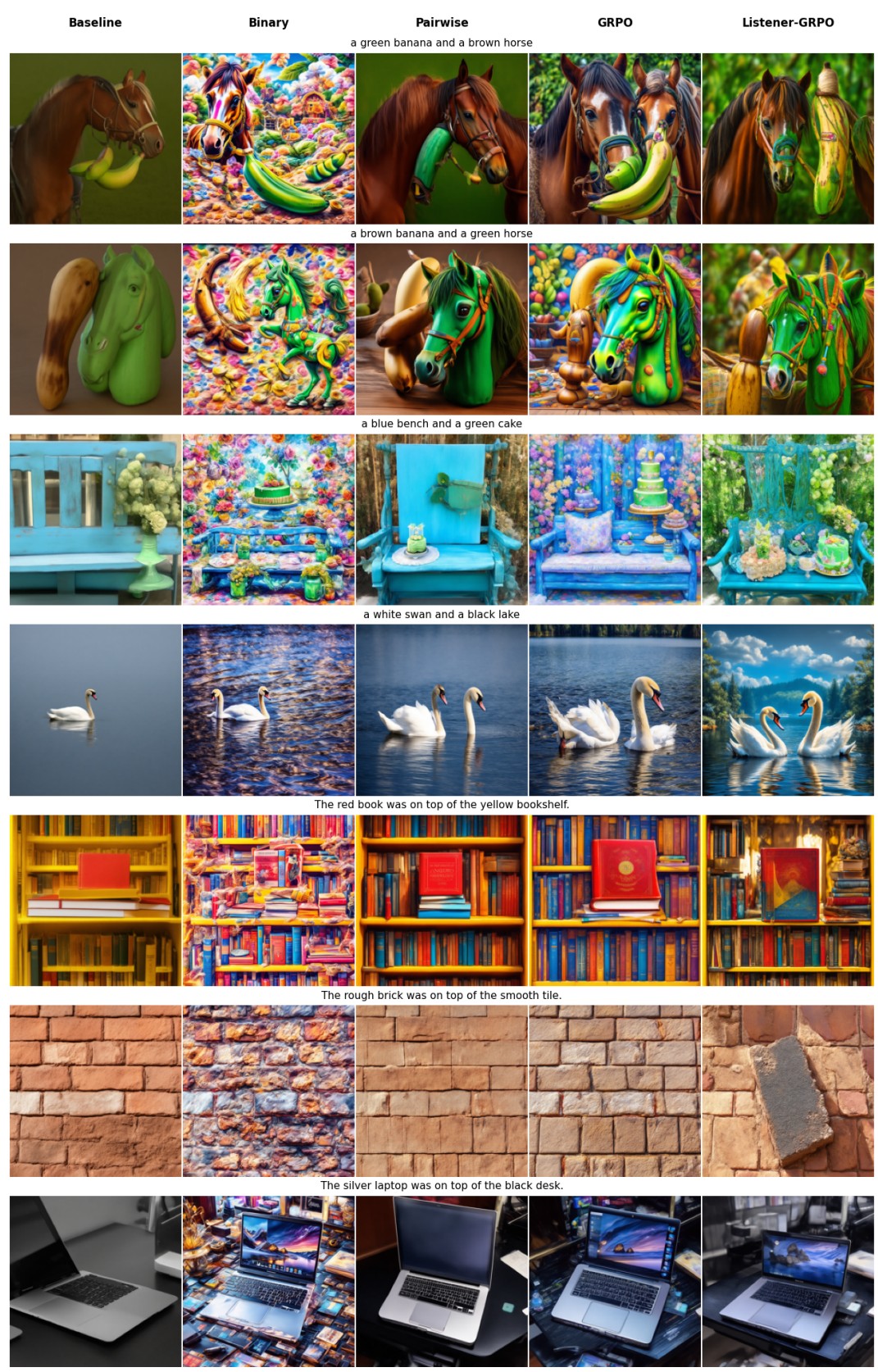

Figure 9: Qualitative comparison of **Stable Diffusion 3.5-Medium** generations. All checkpoints were trained with identical hyperparameters and evaluated with 10 inference steps.

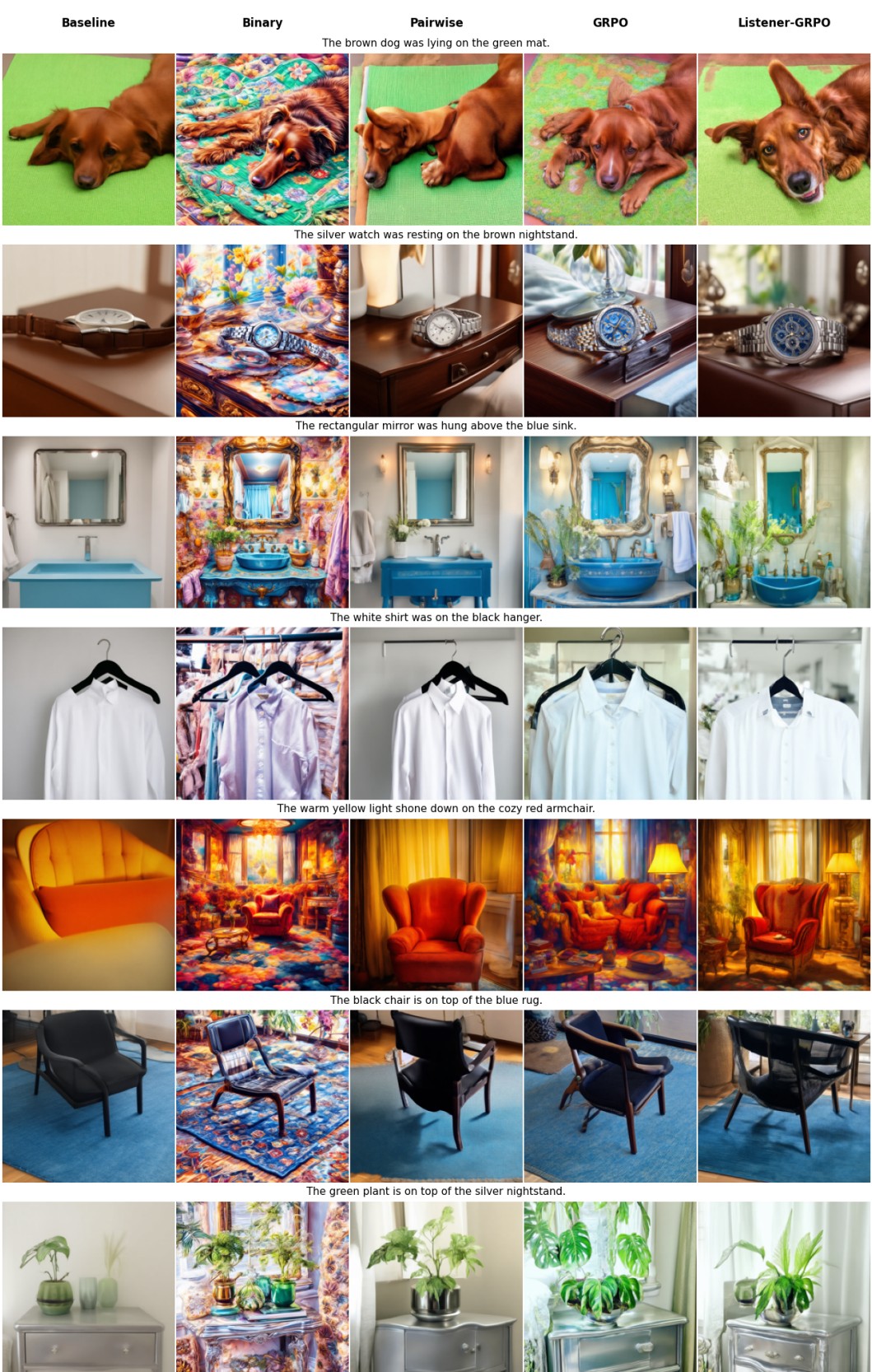

Figure 10: Qualitative comparison of **Stable Diffusion 3.5-Medium** generations. All checkpoints were trained with identical hyperparameters and evaluated with 10 inference steps.

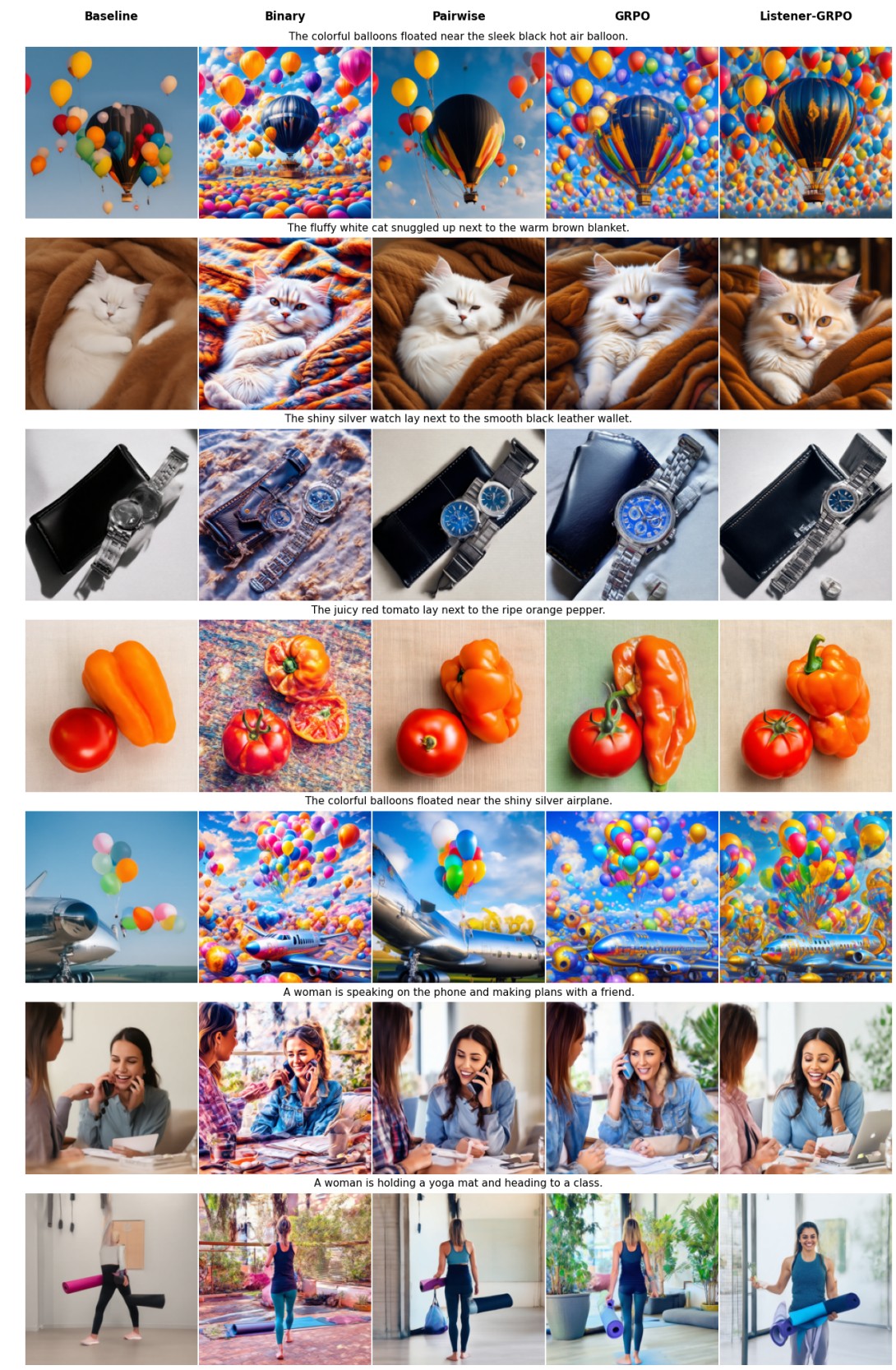

Figure 11: Qualitative comparison of **Stable Diffusion 3.5-Medium** generations. All checkpoints were trained with identical hyperparameters and evaluated with 10 inference steps.

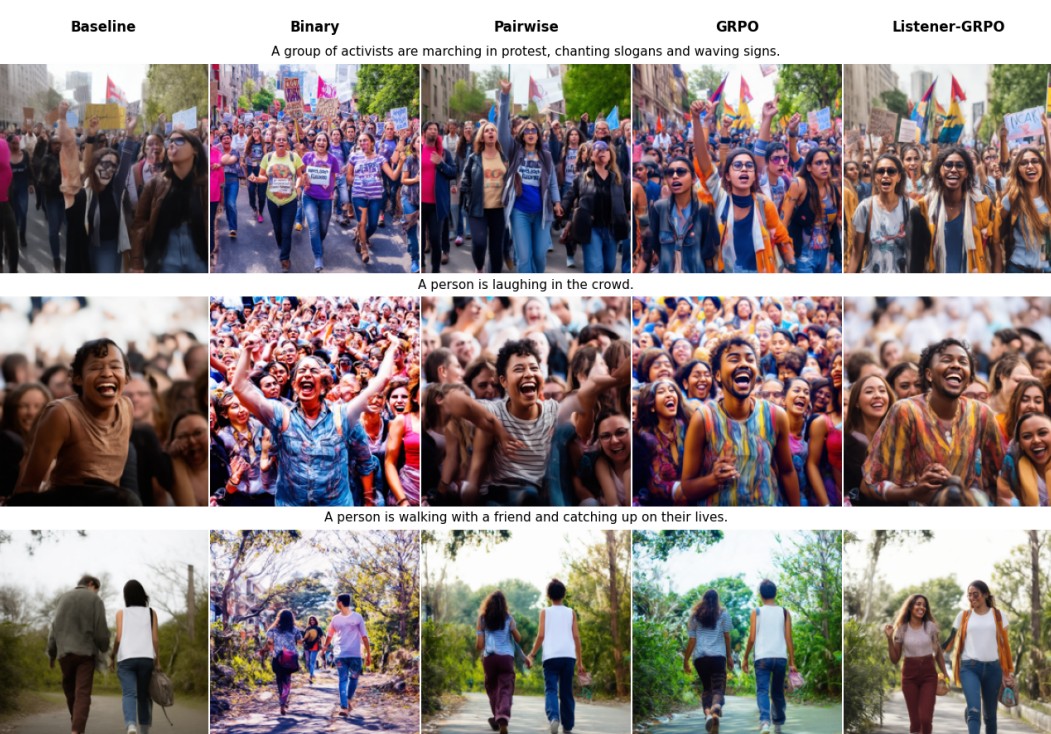

Figure 12: Qualitative comparison of **Stable Diffusion 3.5-Medium** generations. All checkpoints were trained with identical hyperparameters and evaluated with 10 inference steps.

