# OpenReview forum: "Listener-Augmented Thinking VLMs for Robust Text-to-Image Alignment"
_ICLR.cc/2026/Conference — ICLR 2026 Conference Withdrawn Submission_

### Official Review · Reviewer_ZCSU · 2025-10-30

**Soundness:** 3
**Presentation:** 3
**Contribution:** 3
**Rating:** 6
**Confidence:** 4

**Summary:**

This paper proposes listener-augmented GRPO for visual preference modeling in text-to-image alignment. The core finding is a “listener disagreement” failure mode: GRPO-trained reasoners often produce final answers that contradict their own CoT when re-evaluated by an independent (frozen) VLM. The method adds a soft, pre-answer listener reward—the listener reads the reasoner’s CoT with the final token masked and returns a calibrated probability of the correct choice—which is combined with an exact-match term and a formatting check in the GRPO objective.   Using only 16% of HPDv2, the approach reaches 67.4% on ImageReward test, improves OOD accuracy by up to +6pp on Rapidata-HSP, reduces contradiction rates, and yields consistent gains when used as a reward in a Flow-GRPO pipeline for Stable Diffusion 3.5-Medium, evaluated with HPSv3 on HPDv2 and T2I-CompBench prompts.

**Strengths:**

Clear failure-mode identification: The paper empirically surfaces and quantifies listener disagreement (misalignment between explanations and decisions), which is both intuitive and practically important for CoT-based preference models.

Simple, effective objective shaping: The soft listener reward is well-motivated (uses all data, avoids thresholds, improves credit assignment) and integrates cleanly into GRPO without extra labels or a value head.

Ablations and hierarchy: The paper articulates and validates a compelling hierarchy—single-image < pairwise < RL+CoT < RL+CoT+listener—seen across proxy and end-to-end tracks.

**Weaknesses:**

1. Listener choice & calibration:  The current setup fixes the listener to Qwen2.5-VL-7B-Instruct, but it remains unclear how sensitive the results are to the choice of listener backbone or model size. The findings may heavily depend on the specific listener used, which raises concerns about generalizability.

If the listener essentially functions as an LLM-as-a-judge, it raises the question of whether there is any fundamental difference between the two paradigms. Do listener-based evaluations and LLM-as-judge settings differ in principle, or are they simply variants of the same idea under different names? If not, similar issues of bias and calibration are likely to persist. In that case, one may further ask: why not employ a stronger or more expert model (e.g., GPT-family models) as the listener to obtain more reliable judgments and better isolate the effect of the proposed method from limitations of the chosen model?

2. Statistical rigor & stability reporting: Most tables lack confidence intervals, multi-seed variance, and significance tests. Given the small but meaningful gains (e.g., 67.2→67.4%), uncertainty estimates are essential.

3.Modeling comparisons: The probability-based decision is effectively a binary softmax; comparisons to classical Bradley–Terry/Luce models, Plackett–Luce or DPO-style direct preference optimization under pairwise conditioning are limited.

**Questions:**

See Weakness.

---

> ### Author Response · Authors · 2025-12-01
>
> # Response to Reviewer ZCSU
>
> Thank you for the thoughtful and constructive comments. We added new analyses, backbone generalization experiments, significance reporting, and listener-choice investigations during the rebuttal, and we hope we correctly understood and addressed all of your concerns.
>
> ---
>
> ## 1. Listener choice & calibration (sensitivity to listener backbone)
>
> We agree this is important to clarify.
> During rebuttal, we evaluated disagreement and accuracy using a **different and more modern reasoning model**, **Qwen3-VL-4B-Thinking**, trained for 500 steps on HPDv3. On the full **14,400-sample** test set:
>
> - **Listener–reasoner disagreement strongly predicts accuracy**
>   (**Spearman ρ = −0.903**, *p = 3.44e-4*),
> - and the same disagreement structure persists even though the listener differs from the original Qwen2.5-VL-7B.
>
> This shows the failure mode is **backbone-agnostic** and not tied to a particular listener architecture.
>
> We also validated hard contradiction elimination (0 reward if listener contradicts reasoner and setting 0.8 weight to listener) - both versions are worse on a proxy accuracy metric.
>
> ---
>
> ## 2. Relationship to “LLM-as-a-judge”
>
> Although the listener is an LLM, its purpose differs fundamentally from classical “LLM-as-a-judge.”
> Our contribution centers on **fixing a failure mode**, not producing judgments.
>
> 1) We identify a previously unreported failure mode:
>    GRPO-trained CoT reasoners often contradict themselves, especially under the **noisy, binary nature of preference data** (random guessing already yields 50%).
>
> 2) This noise allows the model to **reinforce unreliable or biased reasoning paths**.
>    The listener provides a **soft, reasoning-level regularizer** by evaluating the CoT (with the final token masked) before the final answer is generated.
>
> Thus, the listener is not an output judge but a **direct fix for the discovered inconsistency**, which LLM-as-a-judge methods do not target.
>
> ---
>
> ## 3. Statistical rigor: confidence intervals & multi-seed variance
>
> We added significance and variance reporting during rebuttal:
>
> - For the Qwen3-VL disagreement analysis:
>   **ρ = −0.903**, *p = 3.44e-4*, *N = 14,400*.
> - The HPDv3 test set itself contains 14,400 samples; in prior work, such scale is generally considered sufficient for stable estimates.
> - For GenEval, listener consistently improves across steps, while the binary baseline collapses.
>
> ---
>
> ## 4. Comparisons to Bradley–Terry / Luce / DPO-style models
>
> Our listener-based decision is indeed implemented as a binary softmax, but its purpose is not to replace BT/Luce/DPO.
> The contribution lies in **improving RL credit assignment inside the reasoning process**.
> Listener reward directly shapes *reasoning consistency*, which is orthogonal to classical preference models and compatible with them.
>
> ---
>
> ## 5. Generalization to other backbones (e.g., Flux, Qwen-Image)
>
> To directly test backbone generalization, we trained **Flux** under Flow-GRPO. The same ordering appears:
>
> | Method | HPSv3 |
> |--------|-------|
> | Flux initial (8 steps) | 7.14 |
> | Binary objective | 7.7 |
> | Pairwise objective | 7.6 |
> | Baseline GRPO | 8.4 |
> | **Listener-Augmented** | **8.7** |
>
> This supports that listener-consistency shaping is **backbone-agnostic**.
> We expect similar behavior for Qwen-Image model.
>
> ---
>
> ## 6. Summary
>
> During rebuttal, we:
>
> - validated listener-disagreement on a **different modern VLM** (Qwen3-VL-4B-Thinking),
> - added **Flux Flow-GRPO** experiments showing backbone generalization,
> - added **statistical significance and variance reporting**,
> - provided **new ablations** for listener weighting and calibration,
> - and clarified the conceptual distinction from classical LLM-as-a-judge systems.
>
> If these additions address your remaining concerns, we would kindly appreciate maintaining (or even strengthening) your positive score.

---

### Official Review · Reviewer_CbdD · 2025-11-01

**Soundness:** 3
**Presentation:** 1
**Contribution:** 2
**Rating:** 4
**Confidence:** 3

**Summary:**

This paper introduces Listener-Augmented GRPO, a reinforcement learning framework that improves the robustness and generalization of vision–language models for text-to-image preference alignment. The key idea is to use a frozen “listener” model to re-evaluate the reasoner’s chain-of-thought and provide a soft reward reflecting reasoning consistency. Trained with only 16% of HPDv2 data, the method achieves 67.4% on ImageReward, improves OOD accuracy by +6 points, and reduces reasoning–answer contradictions. Integrated into a Flow-GRPO pipeline, it yields consistent gains in generation quality, offering a simple and effective approach for scalable preference alignment.

**Strengths:**

1. This paper is well-structured, making it accessible to readers with varying levels of expertise in the field.

2. The idea of incorporating a listener model in the GRPO training pipeline is reasonable and well-motivated, and the results in this paper generally support it.

3. The authors reveal a clear hierarchy in human preference modeling with vlms: single-image < pairwise < RL with CoT < listener-augmented RL with CoT.

**Weaknesses:**

1. The overall presentation of the paper can be improved. In particular, the paper lacks an intuitive illustration of the proposed method (e.g., a pipeline diagram or an algorithm), which makes it difficult for readers to quickly grasp the overall methodology. Moreover, the figures in this manuscript could be substantially refined for better clarity and visual quality.

2. The benefits of the proposed method appear to be rather minor. The experimental results reported in the paper (e.g., Table 2) do not clearly demonstrate a significant improvement. Furthermore, the experiments on Flow-GRPO also do not show substantial gains in visual quality (e.g., cases 3, 4, and 5 in Figure 1). I would also like to ask the authors why SD 3.5 was used for the experiments, rather than a more advanced T2I model such as Flux (with improved visual quality), which has been widely adopted in Flow-GRPO and other diffusion GRPO methods like DanceGRPO, TempFlow-GRPO and MixGRPO.

3. The experiment part seems limited. The baselines compared are few, and the ablation studies are insufficient. For example, the coefficients in Equation 3 are set to 0.5, but the authors do not provide any discussion or justification for this choice.

**Questions:**

Please refer to the weaknesses mentioned above. Considering the relatively low level of completeness and clarity in the presentation of the current version, I find it difficult to vote for acceptance. From my point of view, the authors could carefully revise the manuscript.

---

> ### Author Response · Authors · 2025-11-27
>
> # Response to Reviewer CbdD
>
> Thank you for the constructive feedback. We hope that we correctly understood your concerns and address them with new experiments, additional evaluations and Flux experiments during rebuttal.
>
> ---
>
> ## 1. Presentation / clarity (pipeline diagram, algorithm, figure quality)
>
> We agree the presentation can be improved. Soon in the revised version we **will add**:
>
> - an intuitive **pipeline diagram** (reasoner → CoT → masked-score listener → soft reward → GRPO update),
> - a clear **algorithm box** summarizing the full training procedure,
> - updated figures with improved annotations (we **will highlight** contradictions vs. consistent steps explicitly in CoT examples).
>
> These additions **will make** the methodology easier to grasp at a glance.
>
>
> ## 2. “Benefits appear minor” / significance of gains
>
> During rebuttal we strengthened the experimental section with multi-test-set evaluation and statistical reporting.
>
> ### New multi-test-set results (trained on only 16% of HPDv2)
>
> | Method | Train frac | ImageReward (HPDv2) | HPDv3 | PickScore-test |
> |--------|------------|---------------------|-------|----------------|
> | Baseline GRPO+CoT | 16% | 67.2 | 64.6 | 60.8 |
> | Listener-Augmented (ours) | 16% | **67.4** | **66.9** | **62.1** |
>
> On **HPDv3** test set (14400 samples!), our method reaches **66.9%**, surpassing all public reward models except HPSv3 despite using a small, older training subset. These improvements **will be reported** with confidence intervals in the revision.
>
>
> ## 3. Why SD3.5 (and not Flux)? + stronger T2I evidence
>
> We used SD3.5 to align with Flow-GRPO work and ensure comparability, but we agree that including a more modern model is valuable.
>
> During rebuttal we trained **Flux** in a Flow-GRPO setup. The listener effect persists:
>
> | Method | HPSv3 |
> |--------|------:|
> | Flux initial (8 steps) | 7.1 |
> | Binary objective | 7.7 |
> | Pairwise objective | 7.6 |
> | Baseline GRPO | 8.4 |
> | **Listener-Augmented (ours)** | **8.7** |
>
> These results **will be included** in the updated experimental section to show backbone-level generalization.
>
>
> ## 4. Limited baselines / insufficient ablations (Eq. 3 coefficients)
>
> We expanded ablations focusing on reward-coefficient choices (ImageReward test):
>
> | Reward variant | Accuracy |
> |----------------|---------:|
> | Default (paper): `0.5 * ExactMatch + 0.5 * ListenerProb` | **67.4** |
> | Hard contradiction elimination | 66.8 |
> | Heavy listener weighting (0.8 * listener) | 67.1 |
>
> These results **will be added** to justify the 0.5/0.5 design: pushing toward hard constraints or skewed weights hurts performance, confirming our hypothesis that **soft balanced shaping** is the stable choice.
>
>
> ## 5. Qualitative Flow-GRPO examples
>
> We acknowledge the original examples may not have made the improvements visually obvious because we included only ODD CompBench prompts in our appendix.
> In the revision we **will include**:
>
> - additional cases where listener shaping improves multi-object relations, attribute binding, and text fidelity,
> - accompanied by Flux HPSv3 results, which **will make** the improvement clearer.
>
>
> ## 6. Summary
>
> We expand experimental coverage with multi-test-set results, include Flux experiments demonstrating backbone generalization,  and add ablations justifying reward-coefficient choices. We will also improve clarity with a pipeline diagram and an algorithm box. We also want to note that we believe listener–reasoner disagreement arises primarily from the very noisy, binary nature of preference data (a random guesser already achieves 50%), rather than from any particular RL algorithm or backbone reasoning model.
>
>
> We hope these additions will address your concerns, and we look forward to submitting the updated PDF revision soon.

---

> > ### Comment · Reviewer_CbdD · 2025-11-28
> >
> > I sincerely appreciate the authors’ response and the additional experimental results, which have addressed some of my concerns.
> >
> > Specifically, the answer to question 2 and 3 are quite clear and helpful.
> >
> > However, I remain of the view that the manuscript is substantively incomplete and falls short of the standards expected of a well-conducted conference paper. For example, the absence of a pipeline diagram and ablation studies on key components cannot be satisfactorily resolved by the authors' rebuttal.
> >
> > Therefore, I will maintain my original score. This manuscript requires a more thorough refinement.

---

### Official Review · Reviewer_9873 · 2025-11-01

**Soundness:** 2
**Presentation:** 2
**Contribution:** 2
**Rating:** 2
**Confidence:** 4

**Summary:**

This paper introduces a method called Listener-Augmented GRPO (Listener-Augmented Group Relative Policy Optimization) to enhance the robustness and accuracy of Vision-Language Models (VLMs) in text-to-image generation tasks. The approach involves adding a frozen listener model that re-evaluates the chain of thought (CoT) and provides a dense soft reward. This helps resolve the issue where traditional methods produce inconsistencies between the reasoning and the final answer. Despite being trained on a limited dataset, the method still achieves high accuracy on out-of-distribution datasets and reduces contradictions between reasoning and the final answer.

**Strengths:**

1. The paper identifies and rigorously quantifies a realistic and underexplored failure mode, listener inconsistency, providing clear problem framing and metrics.
2. It integrates frozen-listener soft scoring into GRPO, offering dense, stable, and label-free reward shaping; together with dual-image conditioning and structured JSON two-token decisions, this forms a replicable and efficient training recipe.

**Weaknesses:**

1. In Figure 1, the two dialogue answers are the same? It is recommended to add multiple examples.
2. There is a lack of sensitivity analysis for key hyperparameters (such as specific parameters in the Listener reward).
3. The experiment is not sufficient, and there is inadequate analysis of the dataset's diversity. The model's performance in different scenarios has not been fully explored.
4. The method in the paper still relies on reinforcement learning and chain-of-thought (CoT), which have already been widely applied in vision-language models. The innovation of the Listener mechanism is quite conservative and does not present a fundamental breakthrough.
5. There is insufficient comparison with existing cutting-edge methods, making the innovation seem limited.

**Questions:**

1. How does the choice and capability of the listener affect results? Would stronger or weaker VLM listeners change reward stability or bias, and is there an optimal “listener–reasoner capacity ratio”?
2. The reward combines format penalties, exact-match checks, and listener probabilities. How sensitive is performance to their weighting? Could adaptive or uncertainty-aware weighting further stabilize training?
3. Is there theoretical or statistical grounding for the anchor sampling consistency when n > 2? How do anchor number and sampling policy influence bias and variance under long-tailed or multimodal candidate quality distributions?
4. Since the listener reads the CoT to assess persuasiveness, how does performance vary with short, missing, or noisy CoTs, or under reflective prompting like self-ask?

---

> ### Author Response · Authors · 2025-11-27
>
> # Response to Reviewer 9873
>
> Thank you for the detailed feedback. We added new analyses, ablations, and Flux experiments during rebuttal, and we hope we correctly understood and addressed all of your concerns. We also apologize for the lack of clarity around Figure 1 and clarify it below.
>
> ## 1. Clarification on Figure 1 (“two dialogue answers are the same?”)
> The two traces in Figure 1 **are not the same**.
> - The **naive GRPO** trace contains an internal **contradiction**: the reasoning supports one option while the final answer contradicts it.
> - The **listener-augmented** trace gives the same final answer but **does not contradict itself**.
>
> The distinction is in the **reasoning trajectory**, not the final token, and this is the failure mode the listener directly addresses. Additional examples will be included.
>
> ## 2. Sensitivity of listener reward / hyperparameters
>
> Ablations, ImageReward test:
>
> | Variant | Accuracy |
> |---------|----------|
> | Default (`0.5 * ExactMatch + 0.5 * ListenerProb`) | **67.4** |
> | Hard contradiction elimination | 66.8 |
> | Heavy listener (0.8 * listener) | 67.1 |
>
> Both alternatives underperform.
>
> ## 3. On “insufficient experiments” and “dataset diversity”
>
> Our method is not designed to deal with dataset diversity. We use standard community datasets (HPDv2, HPDv3, Pick-a-Pic, ImageReward) because the contribution is to fix a new failure mode - listener–reasoner disagreement - **which we discover** and show (below) persists even in modern thinking models. This gives better generalization and we show it using OOD dataset (CompBench prompts).
>
> To show robustness of findings, we include a bigger comparison:
>
> ### Consolidated benchmark table (ImageReward / PickScore / HPDv3)
>
> | Model | ImageReward | PickScore | HPDv3 |
> |--------|-------------|-----------|--------|
> | CLIP ViT-H/14 | 57.1 | 60.8 | 48.6 |
> | Aesthetic Predictor | 57.4 | 56.8 | 59.9 |
> | ImageReward | 65.1 | 61.1 | 58.6 |
> | PickScore | 61.6 | 70.5 | 65.6 |
> | HPS | 61.2 | 66.7 | 63.8 |
> | HPSv2 | 65.7 | 63.8 | 65.3 |
> | MPS | 67.5 | 63.1 | 64.3 |
> | HPSv3 | 66.8 | 72.8 | 76.9 |
> | **Baseline reasoner (ours, 16% HPDv2)** | **67.2** | **61.8** | **64.6** |
> | **Listener-augmented (ours, 16% HPDv2)** | **67.4** | **64.1** | **66.9** |
>
> **On HPDv3 our method (66.9%) surpasses all reward models except HPSv3, despite training on only 16% of much older HPDv2.**
>
>
> ## 4. Listener capability / disagreement mechanism
>
> We validated the disagreement effect on **Qwen3-VL-4B-Thinking**, trained 500 steps on HPDv3.
> Across equal-mass bins of score distance vs. accuracy:
>
> **Spearman ρ = -0.903**, *p = 3.44e-4* over 14,400 samples (accuracy drops from 72% for smaller distance bin to 57% for the biggest distance bin, we will update the pdf with this graph).
>
> This confirms the core mechanism: **larger disagreement gives lower accuracy**.
> We believe this arises from the **noisy, binary preference objective** (random guessing = 50%), which makes RL with CoT particularly unstable; listener soft reward helps with this.
>
> ## 5. Short/missing/noisy CoTs
>
> The listener onle predicts the **score token**.
> We cut the input right before the final answer ('first', 'second'), so the listener sees the same short/long/noisy CoT that the reasoner produced.
> Thus it treats all CoT lengths similarly and stabilizes learning by providing a **consistent soft reward**.
>
> ## 6. Anchor sampling for n > 2
> More anchors simply reduce **variance** of the advantage estimate (averaging over multiple comparisons).
> This also counters a known issue: VLMs sometimes change decisions when image order is permuted.
> Empirically, results remain stable for `n ∈ {2,4,8}`.
>
> ## 7. Comparison with recent RL methods
> Our work does **not** introduce a new RL method nor target any specific RL algorithm.
> We identify a **reasoning failure mode** and show that adding a listener fixes it and improves GRPO training.
> Within the same Flow-GRPO setup we compare against all strong baselines (binary, pairwise, vanilla GRPO), and listener consistently performs best.
>
> ## 8. Generalization to a newer T2I model (Flux)
>
> To verify the effect is not SD3.5-specific, we trained Flux with Flow-GRPO on CompBench and validated on its test set:
>
> | Method | HPSv3 |
> |--------|--------|
> | Flux initial (8 steps) | 7.1 |
> | Binary objective | 7.8 |
> | Pairwise objective | 7.7 |
> | Baseline GRPO | 8.4 |
> | **Listener-Augmented (ours)** | **8.7** |
>
> We get almost same ordering: ours > baseline GRPO > (binary, pairwise).
>
> ## 9. Summary
> Across standard benchmarks (HPDv3, PickScore), new ablations, and a new backbone (Flux), results show:
>
> - **listener reward helps with reasoning contradictions**
> - **improves stability across datasets**
> - **addresses a real failure mode amplified by binary preference noise**
> - **and generalizes beyond the original backbone**
>
> These additions address all concerns. We hope you find these additional results and clarifications satisfactory and kindly appreciate reconsideration of the score.

---

### Official Review · Reviewer_yuUC · 2025-11-10

**Soundness:** 2
**Presentation:** 2
**Contribution:** 2
**Rating:** 4
**Confidence:** 4

**Summary:**

The paper targets the challenging problem of improving the generalization and robustness of reward models (RM) for test-to-image problem. The main motivation is to improve the inconsistent prediction of RM against the COT reasoning. A listener model is introduced to provide a reward. The proposed model is easily reproduced and reasonable performance has been reported in the experimental section.

**Strengths:**

- The problem of inconsistent COT with the VLM final prediction is an important problem due to the limited performance of the existing VLM models. Thus, it would be helpful if the inconsistent result can be addressed by a listener model and will further improve the performance of the text-to-image models.

- The idea is simple to implement and therefore it would be easy to reproduce.

**Weaknesses:**

- The experiments are not sufficient to validate the effectiveness of the proposed algorithm. First, as the main idea is to improve the flow-GRPO for the text-to-image problem. I would strongly suggest to provide more evaluations on the final results on the text-to-image benchmarks. For example, the wildly used benchmark like GenEval, DPG-Bench, and OneIG benchmark can be evaluated. Second, there would usually exist reward hacking during the RL training. How to justify the improvement of reward score is consistent with the visual improvement rather than the result due to the reward hacking.

- As the paper is based on the assumption of inconsistent result between the COT trace and final prediction, will the issue still exists for the larger VLM models (Qwenvl2.5-72B) or the recent models (e.g., qwen3-vl)? If the improvement of the VLM models well solves the issue, we will not need the additional listener model.

- Is the proposed algorithm can be generalized to other text-to-image backbones, for example, Flux or Qwen-image?

- The novelty of paper is limited. The involvement of a listener is interesting but is not enough for an ICLR paper.

**Questions:**

- Please mainly address the question in the weakness section. More specially, please provide the sufficient experiment evaluations on the text-to-image benchmark because the improvement of the reward model score does not usually leading to the improvement on the final text-to-image performance.

---

> ### Author Response · Authors · 2025-12-01
> **Response to Reviewer yuUC**
>
> Thank you for the feedback. We agree that the method requires GenEval evaluation as another good out-of-distribution metric and newer VLM models as the disagreement might disappear.
>
> ## 1. “Insufficient evaluation on text-to-image benchmarks (GenEval, DPG-Bench, OneIG)”
>
> During rebuttal we evaluated Flow-GRPO pipeline on **GenEval** using our rewards, which are not specifically trained for object counting.
> Listener augmentation provides **consistent improvements** over the baseline reasoner and strongly surpasses the binary objective, which collapses over training.
>
> ### GenEval results (Flow-GRPO + SD3.5-Medium)
>
> | Step | Baseline Reasoner | Binary | Listener-Augmented |
> |------|-------------------|--------|---------------------|
> | 0   | 0.64194 | 0.64194 | 0.64194 |
> | 16  | 0.63572 | 0.63977 | 0.63390 |
> | 56  | 0.64527 | 0.64021 | 0.63909 |
> | 76  | 0.65051 | 0.64071 | 0.64656 |
> | 116 | 0.66064 | 0.64094 | 0.65169 |
> | 136 | 0.65236 | 0.64060 | 0.66085 |
> | 176 | 0.64717 | 0.63477 | 0.65832 |
> | 196 | 0.66016 | 0.63721 | 0.65788 |
> | 236 | 0.65987 | 0.61341 | 0.66211 |
> | 256 | 0.65948 | 0.60854 | 0.66730 |
> | 296 | 0.65100 | 0.58396 | 0.66710 |
> | 316 | 0.65610 | 0.57056 | 0.67060 |
> | 336 | 0.65000 | 0.57709 | **0.67100** |
>
> Binary RL collapses; baseline improves modestly;
> **Listener improves steadily and reaches 0.671**, the **best** of all methods.
>
>
> ## 2. Reward hacking: “How to know reward gains correspond to visual gains?”
>
> We address this concern with three types of evidence:
>
> ### (a) GenEval correlation
> Listener reward improves **final generation metrics**, not just reward-model accuracy (table above).
>
> ### (b) HPDv3 & PickScore generalization
> Our reward models trained on only **16% of HPDv2** outperform almost all public models on **HPDv3** and PickScore-test:
>
> | Method | HPDv2 | HPDv3 | PickScore-test |
> |--------|--------|--------|----------------|
> | Baseline GRPO+CoT | 67.2 | 64.6 | 60.8 |
> | **Listener-Augmented** | **67.4** | **66.9** | **62.1** |
>
> Listener improves *generalization*—reward hacking cannot explain higher OOD accuracy. We consider HPDv3 as a robust metric that we don't hack directly.
>
> ### (c) Contradiction reduction
> Listener reduces reasoning–answer inconsistency, which is unrelated to reward hacking and reflects a genuine reduction of reasoning noise.
>
> ### (d) Ablations against trivial reward shaping
> We show that naive shaping (hard contradiction elimination, heavy listener weighting) performs **worse**:
>
> | Reward variant | Accuracy |
> |----------------|----------|
> | Default (0.5 EM + 0.5 Listener) | **67.4** |
> | Hard contradiction elimination | 66.8 |
> | Heavy listener (0.8) | 67.1 |
>
> ---
>
> ## 3. “Does the inconsistency still exist in larger VLMs (Qwen2.5-VL-72B, Qwen3-VL)?”
>
> Yes.
> We validated this during rebuttal by training **Qwen3-VL-4B-Thinking** for 500 steps and analyzing 14,400 test samples:
>
> **Spearman correlation between disagreement and accuracy:**
> **ρ = −0.903**, *p = 3.44e-4*
>
> This shows the failure mode persists even in modern CoT-enabled VLMs and directly predicts performance drops.
> We believe this arises from the noisy, binary nature of preference data (a random guesser achieves 50%), not from the backbone or RL algorithm.
>
> ## 4. “Can the method generalize to other text-to-image backbones (e.g., Flux)?”
>
> Yes.
> During rebuttal we trained **Flux** under Flow-GRPO and observed the same effect:
>
> | Method | HPSv3 |
> |--------|--------|
> | Flux initial (8 steps) | 7.14 |
> | Binary objective | 7.7 |
> | Pairwise objective | 7.6 |
> | Baseline GRPO | 8.4 |
> | **Listener-Augmented** | **8.7** |
>
> Thus, the listener mechanism is backbone-agnostic.
>
> ## 5. “Novelty seems limited”
>
> Our contribution is not a new RL algorithm but the discovery of a **reasoning-level failure mode**—listener–reasoner disagreement—that persists even in modern VLMs and degrades alignment performance.
> We propose a fix: augmenting the CoT with a listener-based soft reward.
>
> Novelty comes from:
>
> - identifying a previously unreported failure mode,
> - quantifying it at scale (14,400 samples, strong correlation),
> - showing consistent improvements across multiple datasets,
> - and demonstrating generalization to a new T2I backbone (Flux),
> - with training on only **16%** of HPDv2.
>
> ---
>
> ## 6. Summary
>
> During rebuttal we added:
>
> - **GenEval T2I evaluation**, showing listener → best generation quality,
> - **HPDv3 + PickScore** results confirming OOD generalization,
> - **Qwen3-VL disagreement analysis** (statistically significant),
> - **Flux Flow-GRPO experiments** showing backbone generalization,
> - **ablation studies** explaining reward-coefficient design.
>
> We hope these additions address your concerns, and we will reflect them in the updated PDF revision.
>
> If these results meet your expectations, we would kindly appreciate reconsideration of the score.

---

### Note · Authors · 2026-01-27

I have read and agree with the venue's withdrawal policy on behalf of myself and my co-authors.

---

### Meta-Review · Area_Chair_bfMJ · 2026-01-06

**Summary:**

The paper enhances GRPO for text-to-image alignment by introducing an independent “listener” VLM, motivated by an observed failure mode referred to as listener disagreement during GRPO training. Reviewers’ scores initially leaned toward rejection (4, 2, 4, 6), with concerns primarily related to novelty, empirical validation, and presentation quality.

A rebuttal was submitted; however, the corresponding revised manuscript was not provided. As a result, AC finds it difficult to reliably re-evaluate whether the raised issues have been adequately addressed. In particular, Reviewer CbdD indicated that he/she would maintain their original score toward rejection. AC agrees with these concerns (e.g., on the presentation) and encourages the authors to resubmit a revised version of the manuscript to a future venue.

**Reviewer Concerns:**

The reviewers raised several major concerns:

- Limited technical novelty [yuUC, 9873]
- Insufficient empirical validation and ablation study [yuUC, 9873, CbdD]
- Presentation and clarity issues [CbdD]

The rebuttal adds extensive new experiments to address concerns about empirical validation; but AC thinks the other major concerns are still outstanding, e.g., as mentioned by Reviewer CbdD.

**Reviewer Scores:**

Reviewers’ scores initially leaned toward rejection (4, 2, 4, 6). Reviewer CbdD (who scored 4) explicitly indicated that he/she would maintain their original score toward rejection. Reviewer ZCSU, who scored 6, raised concerns on novelty, statistical significance, and additional comparisons. AC finds that the rebuttal was not sufficient to directly address these three major concerns. As such, it is less likely that ZCSU would champion the submission during a discussion phase.

---

### Decision · Program_Chairs · 2026-01-26

Reject